# Association between Depressive Symptoms and Food Insecurity among Indonesian Adults: Results from the 2007–2014 Indonesia Family Life Survey

**DOI:** 10.3390/nu11123026

**Published:** 2019-12-11

**Authors:** Emyr Reisha Isaura, Yang-Ching Chen, Annis Catur Adi, Hsien-Yu Fan, Chung-Yi Li, Shwu-Huey Yang

**Affiliations:** 1Department of Nutrition, Faculty of Public Health, Airlangga University, Surabaya, East Java 60115, Indonesia; emyr.reisha@fkm.unair.ac.id (E.R.I.); annis_catur@fkm.unair.ac.id (A.C.A.); cyli99@mail.ncku.edu.tw (C.-Y.L.); 2School of Nutrition and Health Sciences, College of Nutrition, Taipei Medical University, Taipei 11031, Taiwan; melisa26@gmail.com; 3Department of Family Medicine, School of Medicine, College of Medicine, Taipei Medical University, Taipei 11031, Taiwan; 4Department of Family Medicine, Taipei Medical University Hospital, Taipei 11031, Taiwan; a1118148@ulive.pccu.edu.tw; 5Research Center of Food Science and Technology, Rumah Inovasi Natura, Surabaya, East Java 60112, Indonesia; 6Department and Graduate Institute of Public Health, College of Medicine, National Cheng Kung University, Tainan 70101, Taiwan; 7Nutrition Research Center, Taipei Medical University Hospital, Taipei 11031, Taiwan; 8Research Center of Geriatric Nutrition, College of Nutrition, Taipei Medical University, Taipei 11031, Taiwan

**Keywords:** depressive symptoms, food insecurity, nutrition, adults, generalized estimating equation

## Abstract

Background: Depressive symptoms and food insecurity are two of the public health concerns in developing countries. Food insecurity is linked to several chronic diseases, while little is known about the association between food insecurity and depressive symptoms among adults. A person with limited or uncertain availability or access to nutritionally sufficient, socially relevant, and safe foods is defined as a food-insecure person. Materials and methods: Data were obtained from 8613 adults who participated in the Indonesia Family Life Survey (IFLS) in 2007 and 2014. The 10 items of the food frequency questionnaire (FFQ) were used in food consumption score analysis to assess food insecurity based on the concept of the World Food Program (WFP). Depressive symptoms were assessed using 10 items of the self-reported Center for Epidemiologic Studies Depression (CES-D) questionnaire. A linear and multiple logistic regression model with a generalized estimating equation was used to test the hypothesis while accounting for the health behaviors and sociodemographic characteristics. Results: Food consumption score was negatively associated with CES-D 10 score (*β*-coefficients: −9.71 × 10^−3^ to −1.06 × 10^−2^; 95% CIs: −7.46 × 10^−3^ to −1.26 × 10^−2^). The borderline and poor food consumption group was positively associated with the depressive symptoms, both in the unadjusted and adjusted models (exponentiated *β*-coefficients: 1.13 to 1.18; 95% CIs: 1.06 to 1.28). Conclusions: Depressive symptoms were positively significantly associated with food insecurity. Thus, health professionals must be aware of the issue, and should consider health and nutrition programs for adults at risk of food insecurity.

## 1. Introduction

Depression is a public health problem, associated with adverse mental health, such as suicidal ideation and mortality [1]. Depression is defined as a wide range of mental health problems associated with the negative effect presence, low mood, and emotional, cognitive, physical, and behavioral symptoms [2]. Depression is also a pervasive mental disorder globally that affects all ages [3]. In 2012, depression was estimated to affect about 350 million people globally [3]. Further, the global population with depression was estimated to be 4.4% in 2015, while Indonesia’s national prevalence rate for people having depressive symptoms is 3.7% [4]. Depression or depressive symptoms can occur in episodic sequences [2]. Some unwanted life events (e.g., the loss of a loved one or separation in a relationship), or living with poverty, being unemployed, having a physical illness, and drug and alcohol use-related problems, increase the risk of depression or having depressive symptoms [4,5]. Furthermore, an adult who is unemployed or living in poverty is also associated with food insecurity because of the financial resource limits for acquiring food and managing their diet [6]. 

Food insecurity is defined as a condition in which a person has limited or uncertain availability or access to nutritionally adequate, culturally relevant, and safe foods [6]. Moreover, food insecurity has been found to be associated with chronic diseases [7,8]. The former researchers suggested that chronic diseases may be a contributing factor in the association between food insecurity and depression among the elderly [9,10,11]. On the other hand, food-insecure people are prone to consume an energy-dense and less diverse diet, which eventually results in overweight and obesity, and a higher risk of hypertension, diabetes, and cardiovascular diseases [7,8,12,13]. Seligman and Schillinger suggested that there is a trade-off between providing food and buying medicine in the association between food insecurity, chronic diseases, and depressive symptoms [12]. Not only is the association between food insecurity and depression or depressive symptoms rather vague among adults, but both food insecurity and depression or depressive symptoms can also affect people, women in particular, who live in high-income or low–middle-income countries [14]. Some previous studies found that older adults are prone to the food insecurity issue [15,16,17]. However, the previous study reported that adults in their forties were faced the severe food insecurity issues [18]. Therefore, in this study we used different methods and study designs to further explore and evaluate whether specific age groups modified the association between food insecurity and depressive symptoms among Indonesian adults. We used repeated measurement data to assess the association between food insecurity and depressive symptoms in adults, both in all ages and in various age groups. In addition, we observed depression or depressive symptoms as both predictor and outcome, and used different food insecurity assessments.

## 2. Materials and Methods 

### 2.1. Data Source and Respondents

The current study used secondary data from the fourth (2007) and fifth (2014) waves of the Indonesian Family Life Survey (IFLS; referred to hereafter as IFLS4 and IFLS5, respectively). IFLS datasets comprise anonymous data available to researchers based on the guidelines of the RAND Corporation [19,20,21,22,23]. In the 2007 data, the total number of respondents was 29,059, while in 2014, there were 34,464 respondents (aged 0–80+). For this study, we included adult respondents aged 18–65 years old. We included the same respondents from the year 2007 and the year 2014. Respondents who provided dietary, physical activity, anthropometric, sociodemographic, blood pressure, and depressive symptom data were further analyzed. We excluded respondents who were pregnant or breastfeeding, had a disability, or who were diagnosed with cancer to minimize the sampling bias. Further, we included only respondents who had no missing data in both 2007 and 2014. After the inclusion criteria were applied, 8613 respondents were included in this study. For the purposes of this study, the authors additionally categorized respondents’ ages in years, as follows: less than 40, 40–49, 50–59, and more than 60, besides using responses as continuous data. 

### 2.2. Measurement of Food Insecurity

Measurement of food insecurity in this study followed the concept from the World Food Program (WFP). In practical terms, the definition of food security is related to the failure of the individual to fulfill their need for a nutritious diet [24] in terms of the frequency and diversity of food [25]. Based on the WFP concept, first, we conducted food consumption analysis, resulting in food consumption scores [26,27,28]. We used the same 10 types of food from the food frequency questionnaire (FFQ) in the IFLS4 and IFLS5 questionnaire for the food consumption analysis. The current study used the number of days in which the 10 food types were eaten by respondents in the seven days prior to the interview [25]. Second, the 10 food types of the IFLS4 and IFLS5 FFQ were then grouped into five food groups. The five food groups were the vegetable group (carrot, green leafy vegetables), fruit group (mango, papaya, banana), protein group (eggs, fish, meat), dairy products, and staple group (sweet potato) [19,21]. Third, a total from each food group, called the food consumption score (FCS), was then categorized based on the cutoffs of three food consumption groups (FCGs). The FCS is continuous data, while the FCG is categorical data from the categorization of the FCS. The three FCGs were “poor” if the FCS value was less than 21, “borderline” if the FCS value ranged from 21 to 35, and “acceptable” if the value was more than 35 [25]. Finally, this study defined food-insecure people as those who were in the “poor and borderline” group of FCGs, while food-secure people were defined as those in the “acceptable” group of FCG [28,29]. 

### 2.3. Measurement of Depressive Symptoms

Depressive symptoms were assessed using the self-reported 10 items of the Center for Epidemiologic Studies Depression (CES-D) questionnaire. The CES-D questionnaire is widely used to assess depressive symptoms in adults [30,31]. The 10 CES-D questionnaire answers were in the form of four scales: “rarely or no (≤1 day)”, “some days (1–2 days)”, “occasionally (3–4 days)”, “most of the time (5–7 days)”. The score of each scale’s answer was from zero (“rarely or no”) to four (“most of the time”). We then summarized the score of the 10 CES-D questionnaire answers with a lowest score of 10, and a highest score of 40. Since the score ranged from 10 to 40, the score was rebased to zero to 30, with the highest score referring to the most depressive symptomatology [30]. Previous research suggested the cutoff point for depression or having depressive symptoms was set to a score of ≥10 [30,32,33]. Therefore, respondents were defined as suffering from depression or having depressive symptoms if their CES-D questionnaire score was ≥10.

### 2.4. Measurement of Covariates

The body mass index (in kg/m^2^) was classified into four groups (<18.5, 18.5–25.0, 25.1–27.0, and >27.0) [34]. In addition, a measurement of waist circumference was used for adults aged ≥40 years. Abdominal obesity was determined by respondents’ waist circumference (WC), >90 cm (men) and >80 cm (women) [35]. Hypertension was defined as systolic blood pressure (SBP) ≥140 mmHg or diastolic blood pressure (DBP) ≥90 mmHg [36]. Trained nurses performed the anthropometric and blood pressure measurements. For blood pressure measurements, the respondents were measured twice, before and during the interview, in the seated position [19].

Physical activity was assessed using the number of days for which respondents undertook three types of physical activity (i.e., vigorous, moderate, and walking) within the last seven days. The authors considered days of doing physical activity as a continuous variable in the analysis. Respondents reported in the self-reported questionnaires whether they engaged in physical activities for at least ten minutes continuously during the last seven days. If respondents said yes, then they were further asked about the number of days they did each type of physical activity.

Sociodemographic characteristics were assessed using categorical data, including smoking habit status, level of education, geographical areas of living, and marital status. The respondents’ smoking habit status was categorized into: never (never had a smoking habit), current smoker (currently has a smoking habit), and former smoker (stopped a smoking habit). The respondents’ level of education was categorized into low (<12 years of school attainment) and high (≥12 years of school attainment).

In addition to the covariate variables, we considered adjusting for respondents’ chronic diseases. Therefore, this study used cardiovascular diseases and type 2 diabetes as an adjustment variable in the model. Respondents answered the self-reported question of whether any paramedics ever informed them that they had type 2 diabetes. The respondents also answered the self-reported question of whether any paramedics ever informed them that they had a stroke/heart attack, coronary heart disease, angina, or other heart problems. The authors defined cardiovascular disease as a combination of heart diseases and stroke events [27,37]. If the respondents reported any of the chronic diseases (i.e., diabetes, cardiovascular diseases), then they were asked when their chronic disease was first diagnosed. 

### 2.5. Statistical Analysis

The current study used secondary data with repeated measurements of the same respondents for the years 2007 and 2014. The respondents’ characteristics were presented as means (standard deviation) for the continuous data and numbers (percentages) for the categorical data. The values between groups were compared using a one-way analysis of variance (ANOVA) for the continuous data, and the Bonferroni post-hoc test or chi-squared test for the categorical data. Further, we combined the two datasets (IFLS4 and IFLS5) in the analysis to test the association among variables. Since the data in this study were repeated measurement data from the same respondents throughout the 7-year follow-up period, the authors used regression models with the generalized estimating equation (GEE) method [38]. The GEE is a statistical approach generally used in the analysis of longitudinal data or repeated measurements [39,40,41,42,43,44], with the primary advantage being that it accounts for the within-adults variation [45]. Firstly, we used a linear regression model with GEE to assess the association between the food consumption score and the CES-D score. The linear regression used the Gaussian distribution (family) for the dependent variables, an identity link function, and “independent” for the correlation matrix. Secondly, we used a binary logistic regression model with GEE to assess the association between food consumption groups and depressive symptoms. The authors used the “acceptable” FCG group as the reference group in the logistic regression model with GEE. The logistic regression model used the “binomial” distribution (family) for the dependent variable, a log link function, and an “independent” correlation matrix. The exponentiated beta coefficient was also estimated from the logistic regression to assess the relationship of interest [45,46]. This study used three models that accounted for various potential confounders in the multiple logistic regression model with GEE. The first model was an unadjusted model and the second model was with an adjustment for age and gender. The third model was with further adjustment for level of education, marital status, geographical areas of living, smoking habit status, physical activity days, blood pressure values, body mass index, and included diabetes and cardiovascular diseases. The last model (model 3) was a full adjustment model. A similar sequence of adjustments for potential confounders was also used for multiple linear regression models with GEE. Statistical significance was set to the *p*-value < 0.05. The post hoc test was conducted to retest the complete adjustment estimation models for every age group category. All the analyses were conducted using STATA statistical software (V 12.1; StataCorp LP, College Station, Texas, TX, USA).

## 3. Results

Table 1 shows the characteristics of the 8613 respondents by food security groups in 2007 and 2014. Respondents included 3999 women and 4614 men. The prevalence rates of food insecurity (borderline and poor) increased from 2007 to 2014. The borderline FCG prevalence rates increased from *n =* 1474 (17.11%) in 2007 to *n =* 2911 (33.80%) in 2014. Meanwhile, the prevalence rates of poor FCG also increased from *n =* 693 (8.05%) in 2007 to *n =* 1713 (19.89%) in 2014. The majority of respondents in this study had a low level of education (less than 12 years of school attainment) in both year 2007 and 2014 (*p* < 0.001). The percentage of food-insecure people living in urban areas increased from 2007 (borderline = 18.87%; poor = 9.55%) to 2014 (borderline = 35.31%; poor = 23.46%) (*p* < 0.001). The percentage of food-insecure people with abdominal obesity increased from 2007 (borderline = 15.74%; poor = 6.81%) to 2014 (borderline = 32.75%; poor = 18.00%) (*p* < 0.001). As shown in Table 1, the number of respondents who had depressive symptoms increased from *n =* 955 in 2007 to *n =* 2616 people in 2014. To compare body mass index (BMI), body shape index, waist circumference, blood pressure, food consumption score, physical activity days, and CES-D score, we used a one-way analysis of variance (ANOVA) with Bonferroni post hoc test. The results of the Bonferroni post hoc test are in Appendix A.

Table 2 presents the overall and age-specific proportions of food consumption groups among people with depressive symptoms. In 2007, the overall (range of age-specific proportion) proportion of acceptable, borderline, and poor FCG was 11.65% (range: 0.13%–52.46%), 9.02% (range: 0.75%–54.89%), and 10.24% (range: 0.00%–57.75%), respectively. In 2014, corresponding figures were 32.09% (range: 10.70%–35.70%), 29.34% (range: 10.66%–33.49%), and 28.14% (range: 10.79%–39.63%), respectively. The prevalence of depressive symptoms significantly varied with age. Except for the borderline group in 2007, the proportion of other food consumption groups in both years also significantly varied. 

Table 3 demonstrates the association between food consumption groups and the depressive symptoms outcomes among adults. The food consumption score was negatively significantly associated with the CES-D score both in the unadjusted model (*β*-Coefficients: −9.51 × 10^−3^ (95% CI: −6.45 × 10^−3^, −1.26 × 10^−2^)) and adjusted models (*β*-Coefficients: −9.71 × 10^−3^ (95% CI: −6.62 × 10^−3^, −1.28 × 10^−2^) to *β*-Coefficients: −1.04 × 10^−2^ (95% CI: −7.26 × 10^−3^, −1.36 × 10^−2^)). Further, we used the logistic models to compare food security as represented by acceptable FCG and food insecurity as represented by borderline and poor FCG. The borderline group was positively associated with the depressive symptoms of both the unadjusted and adjusted models with exponentiated *β*-Coefficients of 1.13 (95% CI: 1.06 to 1.21) to 1.18 (95% CI: 1.10 to 1.26). The depressive symptoms of the borderline group will increase by 1.13–1.18 units for every one-unit increase of the acceptable group. On the other hand, the poor group was also significantly positively associated with the depressive symptoms in both the unadjusted and adjusted models, with exponentiated *β*-Coefficients of 1.17 (95% CI: 1.07 to 1.27) to 1.22 (95% CI: 1.12 to 1.33). The depressive symptoms of the poor group will increase by 1.17–1.22 units for every one-unit increase of the acceptable group. 

Table 4 shows the results of age-specific analyses for the relationship between food insecurity (as represented by FCS and FGC) and depression or depressive symptoms (as represented by the CES-D score). The current study used a full adjustment model (model 3) in the analysis to examine the findings’ post hoc stability and decide whether the regression analysis differed based on the age group. The poor food consumption group was significantly and independently positively associated with depressive symptoms among respondents aged 40–49 years, with an exponentiated *β*-Coefficient of 1.24 (95% CI: 1.08 to 1.42). The depressive symptoms of the poor food consumption group will increase by 1.24 units for every one-unit increase of the acceptable food consumption group only among respondents aged 40–49 years. The remaining age groups did not report a food consumption score nor food consumption groups that were significantly associated with depressive symptoms.

## 4. Discussion

The present study aimed to explore the association between food insecurity and depressive symptoms among adults aged 18–65 years in Indonesia. The borderline and poor food consumption groups represent food insecurity. The present study results suggest that food insecurity was positively significantly associated with depressive symptoms in Indonesian adults. As expected, the secondary findings confirmed that the high prevalence of depressive symptoms occurred among respondents with food insecurity across all ages of adults. Further, the total prevalence rates of food-insecure respondents with depressive symptoms (borderline FCG: 29.3%; poor FCG: 28.1%) was higher than the prevalence rates of food-secure respondents with depressive symptoms (acceptable FCG: 32.1%). The present study’s prevalence rates are higher than the national crude prevalence rate of depressive symptoms, which was 3.7% in 2015 [4]. Therefore, the government, health practitioners, and relevant stakeholders need to be more concerned about the issue of food insecurity and depressive symptoms. 

One possible action that might help is a food insecurity and depressive symptoms’ screening and monitoring process, along with the nutrition health programs for adults. Previous researchers found that the level of education is associated with food insecurity and the increased individual level of stress, which may lead to depressive symptoms [47,48,49]. Another possible reason is people with less education will more likely experience economic hardship, due to a lower-paid work type or unemployment, which is associated with food insecurity and depressive symptoms [50]. The findings in this study were in line with those of previous research, indicating that the majority of food-insecure respondents had a low level of education and lived in urban areas, with a greater associated risk of economic hardship compared to people with a higher level of education [51]. 

Moreover, adults who experience a high-burden work type with less income may have depressive symptoms, which can interfere with the ability to manage financial affairs related to food choice and preparation [52,53]. Furthermore, former researchers suggest that unhealthy food choices, for example, Western dietary patterns, which are more likely to contain high calories, high fat, and less diversity, are (partly) associated with depressive symptoms [54,55]. The food consumption score analysis based on the WFP concept is more concerned with the food frequency and quality, and the diversity of diet [25]. One of the explanations is in the food consumption analysis, in which the calculation of food consumption score includes the number of days during which the respondent eats the food type in the FFQ, multiplied by the weight score of each food group type. The highest score refers to all of the food with relatively high energy, good-quality protein, and micronutrients [28]. Therefore, the higher the food consumption score, the better and more diverse the diet and the less food insecurity. However, the present study found that food-insecure respondents had lower food consumption scores than food-secure respondents, indicating that food-insecure respondents possibly consumed lower quality and less diverse food, with high energy and fat density.

Food insecurity is associated with depressive symptoms, overweight and obesity, hypertension, diabetes, and cardiovascular diseases [56,57,58,59,60,61,62]. The results of the present study were in line with previous research. The respondents in borderline and poor FCG have lower FCS, and higher body mass index, waist circumference, systolic blood pressure, and CES-D score, than the respondents in the acceptable FCG. One of the reasons to explain the mechanism between food insecurity, overweight, hypertension, and depression is when food-insecure people are unable to choose a properly balanced meal for themselves, and thus eat a low-quality and less diverse diet (high energy, high fat), which eventually leads to being overweight [56]. Food-insecure people are not only at a higher risk of being overweight, but also of increased levels of stress, possibly from a lack of sleep quality due to hunger or worries about providing food the next day [52,63]. On the other hand, continuous food insecurity in a person’s life may lead to the onset of depression [11,64]. Pryor and colleagues suggested that food insecurity during young adulthood (18–35 years) co-occurs with three types of mental health problems (i.e., depression, suicidal ideation, and substance use problems in young adulthood) [65]. 

Furthermore, people with food insecurity are more likely to experience depression and undertake less leisure-time physical activity than those with food security [66,67,68,69]. The current study results support the evidence from previous research that the mean of vigorous physical activity (VPA) and moderate physical activity (MPA) days was different between the acceptable FCG (food-secure) and borderline or poor FCG (food-insecure). Moreover, the association between food insecurity and depressive symptoms might be affected by several health factors, which need further exploration using a longitudinal study or more variables. Thus, we further tested the association between food insecurity and depressive symptoms using regression analysis. The results suggested the association was constant even after gradually adjusting for the covariates. The covariates included health and sociodemographic characteristics, such as age, gender, level of education, marital status, geographical area of living, smoking habit status, blood pressure, BMI, incidence of diabetes, and cardiovascular diseases. Taken together, food insecurity was found to have a positive effect on depressive symptoms even after adjustment. The post hoc result showed that respondents aged 40–49 years independently reported levels of poor FCG that were significantly associated with depressive symptoms. The present study results were in line with a previous study that showed that people aged 40–49 are confronted with the most severe problems of food insecurity [18]. The study by Ziliak and Gundersen reported that the “youngest old” suffer from the most severe form of food insecurity compared to those of a younger age or even those over 70 years [70]. The middle-aged food-insecure people might face a recession of income, live in poverty in urban areas, be raising grandchildren, have a limitation on their activities of daily living, or be in a minority [71]. 

There are several limitations in the present study. First, the dataset that we used was restricted to the selected variables (i.e., the use of the food frequency questionnaire to conduct the food insecurity assessment) for the original study because we used secondary data in this study. However, the FFQ used in this study was widely used from the first wave of the Indonesia Family Life Survey, initiated in 1993, and has also been used in several previous studies [72,73,74]. Second, the assessment of food insecurity and depressive symptoms was limited to self-reported data. However, the food insecurity measurement from the FFQ was relevant when we defined it from the food frequency and diversity diet [24,25]. Moreover, the use of the CES-D 10 items is widely used to measure the depression or depressive symptoms among adults [30,75]. Third, we could not control for the respondents who received antidepressants or therapy because the IFLS questionnaire did not include a related question. Thus, we suggest future research should further explore the socio-environmental and dietary risk factors of depression and food insecurity. The present study concerns a very important and, at the same time, complex topic of depressive symptoms and lack of food security. These are two public health problems in developing countries that, along with obesity-related non-communicable diseases, significantly affect people’s quality of life.

## 5. Conclusions

To our knowledge, the present study results contribute to the evidence that food insecurity is positively significantly associated with depression symptoms among South-East Asian, particularly Indonesian, adults, as well as for people aged 40–49. Hence, depressive symptoms and food insecurity are public health concerns that need to be improved by health professionals, in conjunction with health and nutrition programs for adults who are at risk of, or currently experiencing, either of these issues. Health professionals must be aware of screening, monitoring, and engaging with food-insecure adults to prevent depression or chronic diseases.

## Figures and Tables

**Table 1 nutrients-11-03026-t001:** Respondents’ characteristics by food security group.

	2007				2014			
	Acceptable	Borderline	Poor	*p*-Value	Acceptable	Borderline	Poor	*p*-Value
*n* (%)	6446 (74.84)	1474 (17.11)	693 (8.05)		3989 (46.31)	2911 (33.80)	1713 (19.89)	
Age (years), mean (SD)	41 (9)	40 (9)	41 (9)		48 (9)	47 (9)	48 (9)	
Age (years), *n* (%)				0.110				0.004
<40	2946 (74.64)	705 (17.86)	296 (7.50)		918 (45.22)	743 (36.60)	369 (18.18)	
40–59	2210 (75.81)	472 (16.19)	233 (7.99)		1357 (47.58)	949 (33.27)	546 (19.14)	
50–59	1286 (73.70)	295 (16.91)	164 (9.40)		1243 (46.42)	881 (32.90)	554 (20.69)	
≥60	4 (66.67)	2 (33.33)	0 (0.00)		471 (44.73)	338 (32.10)	244 (23.17)	
Sex, *n* (%)				0.021				0.034
Women	2937 (73.44)	721 (18.03)	341 (8.53)		1799 (44.99)	1365 (34.13)	835 (20.88)	
Men	3509 (76.05)	753 (16.32)	352 (7.63)		2190 (47.46)	1546 (33.51)	878 (19.03)	
Level of Education, *n* (%)				<0.001				<0.001
Low (<12 years attainment)	4162 (70.00)	1185 (19.93)	599 (10.07)		2365 (40.27)	2054 (34.97)	1454 (24.76)	
High (≥12 years attainment)	2284 (85.64)	289 (10.84)	94 (3.52)		1624 (59.27)	857 (31.28)	259 (9.45)	
Marital Status, *n* (%)				0.231				0.309
Married or ever married	5954 (74.76)	1358 (17.05)	652 (8.19)		3854 (46.33)	2820 (33.90)	1645 (19.77)	
Single or Never Married	492 (75.81)	116 (17.87)	41 (6.32)		135 (45.92)	91 (30.95)	68 (23.13)	
Geographical areas of living, *n* (%)				<0.001				<0.001
Rural	3050 (71.58)	804 (18.87)	407 (9.55)		1471 (41.23)	1260 (35.31)	837 (23.46)	
Urban	3396 (78.03)	670 (15.40)	286 (6.57)		2518 (49.91)	1651 (32.73)	876 (17.36)	
Smoking Habit Status, *n* (%)				0.124				0.003
Never	3827 (75.22)	864 (16.98)	397 (7.80)		2271 (46.61)	1639 (33.64)	962 (19.75)	
Current Smoker	2461 (73.90)	582 (17.48)	287 (8.62)		1440 (44.65)	1116 (34.60)	669 (20.74)	
Former smoker	158 (81.03)	28 (14.36)	9 (4.62)		278 (53.88)	156 (30.23)	82 (15.89)	
Using Diabetes Medication, *n* (%)				0.622				0.468
No	6437 (74.85)	1472 (17.12)	691 (8.03)		3925 (46.24)	2872 (33.83)	1692 (19.93)	
Yes	9 (69.23)	2 (15.38)	2 (15.38)		64 (51.61)	39 (31.45)	21 (16.94)	
Using Hypertension Medication, *n* (%)				0.173				0.007
No	6391 (74.77)	1468 (17.17)	689 (8.06)		3796 (46.01)	2794 (33.86)	1661 (20.13)	
Yes	55 (84.62)	6 (9.23)	4 (6.15)		193 (53.31)	117 (32.32)	52 (14.36)	
Using Cholesterol Medication, *n* (%)				0.557				0.002
No	6442 (74.85)	1472 (17.10)	692 (8.04)		3906 (46.08)	2876 (33.93)	1695 (20.00)	
Yes	4 (57.14)	2 (28.57)	1 (14.29)		83 (61.03)	35 (25.74)	18 (13.24)	
Abdominal obesity ^†,^ *n* (%)				0.001				<0.001
No	2108 (73.48)	485 (16.90)	276 (9.62)		1556 (44.43)	1158 (33.07)	788 (22.50)	
Yes	1388 (77.46)	282 (15.74)	122 (6.81)		1516 (49.25)	1008 (32.75)	554 (18.00)	
Body Mass Index (kg/m^2^), mean (SD)	23.31 (4.16)	22.86 (4.09)	22.50 (4.07)	<0.001	24.31 (4.33)	24.10 (4.44)	23.42 (4.31)	<0.001
Body Mass Index ^‡^, *n* (%)				<0.001				<0.001
<18.5	562 (70.07)	154 (19.20)	86 (10.72)		285 (41.07)	242 (34.87)	167 (24.06)	
18.5–25.0	3978 (74.02)	943 (17.55)	453 (8.43)		2099 (44.74)	1570 (33.46)	1023 (21.80)	
25.1–27.0	771 (78.19)	147 (14.91)	68 (6.90)		610 (50.12)	412 (33.85)	195 (16.02)	
>27.0	1135 (78.22)	230 (15.85)	86 (5.93)		995 (49.50)	687 (34.18)	328 (16.32)	
Hypertension, *n* (%)				0.716				0.093
No	4485 (75.09)	1014 (16.98)	474 (7.94)		2410 (46.59)	1773 (34.27)	990 (19.14)	
Yes	1961 (74.28)	460 (17.42)	219 (8.30)		1579 (45.90)	1138 (33.08)	723 (21.02)	
Diabetes, *n* (%)				0.182				0.660
No	6431 (74.88)	1468 (17.09)	689 (8.02)		3854 (46.24)	2817 (33.80)	1663 (19.95)	
Yes	15 (60.0)	6 (24.00)	4 (16.00)		135 (48.39)	94 (33.69)	50 (17.92)	
Cardiovascular Disease, *n* (%)				0.097				0.424
No	6390 (74.75)	1467 (17.16)	691 (8.08)		3886 (46.32)	2828 (33.71)	1675 (19.97)	
Yes	56 (86.15)	7 (10.77)	2 (3.08)		103 (45.98)	83 (37.05)	38 (16.96)	
Depression *, *n* (%)				0.011				0.004
No	5695 (74.37)	1341 (17.51)	622 (8.12)		2709 (45.17)	2057 (34.30)	1231 (20.53)	
Yes	751 (78.64)	133 (13.93)	71 (7.43)		1280 (48.93)	854 (32.65)	482 (18.43)	
Body Shape Index (m^11/6^ kg^−2/3^), mean (SD)	0.0814 (0.0056)	0.0816 (0.0059)	0.0815 (0.0056)	0.028	0.0814 (0.0056)	0.0815 (0.0056)	0.0816 (0.0059)	0.726
Waist Circumference (cm), mean (SD)	82.22 (10.86)	80.70 (10.62)	79.10 (10.77)	<0.001	85.29 (11.51)	84.47 (11.50)	82.70 (12.00)	<0.001
Systolic BP (mmHg), mean (SD)	129.72 (19.12)	130.43 (19.82)	130.86 (19.52)	0.184	135.51 (23.07)	136.31 (23.88)	138.09 (23.72)	<0.001
Diastolic BP (mmHg), mean (SD)	81.41 (11.64)	81.49 (11.48)	81.25 (10.65)	0.899	82.93 (13.16)	83.22 (13.33)	83.35 (13.27)	0.467
Food Consumption Score, mean (SD)	60.71 (18.26)	29.32 (3.90)	15.07 (4.90)	<0.001	46.99 (10.68)	29.81 (4.04)	13.69 (5.43)	<0.001
Walking PA Days, mean (SD)	4 (3)	4 (3)	4 (3)	0.264	4 (3)	4 (3)	4 (3)	0.149
Moderate PA Days, mean (SD)	3 (3)	2 (3)	2 (3)	0.001	3 (3)	2 (3)	2 (3)	0.001
Vigorous PA Days, mean (SD)	1 (2)	1 (2)	1 (2)	0.207	1 (2)	1 (2)	1 (2)	0.019
CES-D 10 Score, mean (SD)	6.19 (3.29)	5.77 (3.16)	5.48 (3.58)	<0.001	8.37 (4.81)	7.93 (4.90)	7.59 (5.24)	<0.001

Notes: BP, blood pressures; PA, physical activity; CES-D 10, Center for Epidemiological Studies Depression 10 items; SD, standard deviation. *n* (%) was for categorical data and mean (SD) was for continuous data presentation. † A definition of abdominal obesity was if women had waist circumference >80 cm or men had waist circumference >90 cm. ‡ Body Mass Index used the cutoff values for the Indonesian adults from the Ministry of Health of Indonesia. * Depression = CES-D 10 score ≥10.

**Table 2 nutrients-11-03026-t002:** The food consumption groups co-occurring with depressive symptoms by age among adults.

	All ages	<40	40–59	50–59	≥60	*p*-Value
2007						
Depressive Symptoms, *n* (%)	955 (11.09)	508 (53.19)	294 (30.79)	151 (15.81)	2 (0.21)	<0.001
Food Consumption Groups, *n* (%)						
Acceptable	751 (11.65)	394 (52.46)	235 (31.29)	121 (16.11)	1 (0.13)	0.001
Borderline	133 (9.02)	73 (54.89)	38 (28.57)	21 (15.79)	1 (0.75)	0.059
Poor	71 (10.24)	41 (57.75)	21 (29.58)	9 (12.68)	0 (0.0)	0.014
2014						
Depressive Symptoms, *n* (%)	2616 (30.4)	719 (27.48)	934 (35.70)	683 (26.11)	280 (10.70)	<0.001
Food Consumption Groups, *n* (%)						
Acceptable	1280 (32.09)	349 (27.27)	457 (35.70)	337 (26.33)	137 (10.70)	<0.001
Borderline	854 (29.34)	251 (29.39)	286 (33.49)	226 (26.46)	91 (10.66)	0.003
Poor	482 (28.14)	119 (24.69)	191 (39.63)	120 (24.90)	52 (10.79)	<0.001

Notes: Depressive symptoms were defined as CES-D 10 score ≥10. Prevalence rates are shown as numbers (weighted prevalence). Depression rates between food consumption groups within the age group were significant (*p*-value = 0.004–0.011).

**Table 3 nutrients-11-03026-t003:** The association between food consumption groups and the depressive symptoms outcomes among adults.

Variables	Model 1		Model 2		Model 3	
	*β* (95% CI)	*p*-Value	*β* (95% CI)	*p*-Value	*β* (95% CI)	*p*-Value
FCS	−9.51 × 10^−3^ (−6.45 × 10^−3^, −1.26 × 10^−2^)	<0.001	−9.71 × 10^−3^ (−6.62 × 10^−3^, −1.28 × 10^−2^)	<0.001	−1.06 × 10^−2^ (−7.46 × 10^−3^, −1.38 × 10^−2^)	<0.001
Acceptable	1 (Ref.)		1 (Ref.)		1 (Ref.)	
Borderline*	1.16 (1.08–1.24)	<0.001	1.15 (1.08–1.23)	<0.001	1.13 (1.06–1.21)	<0.001
Poor*	1.18 (1.09–1.28)	<0.001	1.17 (1.08–1.27)	<0.001	1.17 (1.07–1.27)	<0.001

Notes: CI, confidence interval; FCS, food consumption score. FCS is continuous data of the food security assessment. Depressive symptoms were defined as CES-D 10 score ≥10. Model 1: Unadjusted model. Model 2: Model 1 with adjustment for age and gender. Model 3: Model 2 with adjustment for level of education, marital status, geographical areas of living, smoking habit status, physical activity days, blood pressures, body mass index, diabetes, and cardiovascular diseases. * The exponentiated *β-*coefficient was used for the logistic models of generalized estimating equation.

**Table 4 nutrients-11-03026-t004:** The association between food consumption groups and the depressive symptoms outcomes among adults by specific age group.

Variables	<40 years		40–49 years		50–59 years		≥60 years	
	*β* (95% CI)	*p*-Value	*β* (95% CI)	*p*-Value	*β* (95% CI)	*p*-Value	*β* (95% CI)	*p*-Value
FCS	1.65 × 10^−3^ (−5.43 × 10^−3^, 8.73 × 10^−3^)	0.649	−4.27 × 10^−3^ (−9.99 × 10^−3^, 1.46 × 10^−3^)	0.144	1.11 × 10^−3^ (−4.47 × 10^−3^, 6.70 × 10^−3^)	0.696	5.43 × 10^−3^ (−3.63 × 10^−3^, 1.45 × 10^−2^)	0.240
Acceptable	1 (Ref.)		1 (Ref.)		1 (Ref.)		1 (Ref.)	
Borderline*	0.94 (0.83–1.07)	0.354	1.07 (0.95–1.20)	0.269	0.98 (0.85–1.11)	0.711	1.00 (0.81–1.24)	0.964
Poor*	1.00 (0.85–1.17)	0.986	1.24 (1.08–1.42)	0.002	0.87 (0.73–1.03)	0.111	0.79 (0.60–1.03)	0.082

Notes: CI, confidence interval; FCS, food consumption score. FCS is continuous data of the food security assessment. Depressive symptoms were defined as CES-D 10 score ≥10. Models are adjusted for age, gender, level of education, marital status, geographical areas of living, smoking habit status, physical activity days, blood pressures, body mass index, diabetes, and cardiovascular diseases. * The exponentiated *β*-coefficient was used for the logistic models of generalized estimating equation.

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
