# Peer review of "Association between Depressive Symptoms and Food Insecurity among Indonesian Adults: Results from the 2007–2014 Indonesia Family Life Survey"

_nutrients, 2019, doi:10.3390/nu11123026_

Round 1
Reviewer 1 Report
Dear Authors,
Both, depressive symptoms and food insecurity become two of the public health challenges for low-middle income countries. It is therefore important to try to find a correlation between them, what these studies are undertaking. Large samples, repeated after seven years, were interesting material for analysis.
However, there are several areas that need to be addressed to help improve the paper. Please find below the areas for improvement:
Line 131. Which values was compared using t-test, since there were 3 groups (Acceptable, Borderline and Poor)?
Lines 142- 152 It is unclear why 5 versions of the adjusted models were made and why model 6 with all potential confounders was chosen.
Table 3. There is no p-value for the variables. It's definitely better to compare OR instead of beta-coefficients.
Table 4. There is no p-value for the variables. It's definitely better to compare OR instead of beta-coefficients.
In the logistics model, the results should be related to the reference level.
Correlation between depressive symptoms and food insecurit was observed only in the 40-40 years group. This relationship has not been interpreted precisely.
The analysis are only partially reflected in the conclusions.
Author Response
REVIEWER 1
Comment 1:
Dear Authors,
Both depressive symptoms and food insecurity become two of the public health challenges for low-middle income countries. It is therefore important to try to find a correlation between them, what these studies are undertaking. Large samples, repeated after seven years, were interesting material for analysis.
However, there are several areas that need to be addressed to help improve the paper. Please find below the areas for improvement:
Line 131. Which values was compared using t-test, since there were 3 groups (Acceptable, Borderline and Poor)?
Response 1:
Thank you for the constructive comments on the application of t-test on our study. The first attempt of using t-test was to compare between food security and food insecurity group. We found that after some of revision, we have not yet revise that word. Therefore, in the current version of the manuscript, we revised it. The analysis that we used to compare the mean of the three groups was analysis of variance (ANOVA) and we further did the Bonferroni post hoc test.
We had addressed the ANOVA test on page 4 line 157-158. “The values between groups were compared using a one-way ANOVA for the continuous data with the Bonferroni post-hoc test or chi-square test for the categorical data.”
We added some explanation about the post hoc test using table S1 and S2 in the supplemental materials, and in the results section on page 5-6 line 195-257.
“To compare body mass index, body shape index, waist circumferences, blood pressures, food consumption score, physical activity days, and CES-D score, we used a one-way analysis of variance (ANOVA) with Bonferroni post hoc test. In 2014, the mean body mass index (BMI) was significantly difference between the food consumption groups [F (2, 8610) = 25.12, p < 0.001]. Post hoc comparisons using the Bonferroni test were carried out. BMI in the acceptable group was significantly different from the BMI in the borderline group. BMI in the acceptable group was also significantly different from the BMI in the "poor" group (p < 0.001). However, the BMI of the borderline group did not significantly differ from the BMI of the "poor" group. The mean waist circumference (WSC) was significantly different between the food consumption groups [F (2, 4658) = 23.38, p < 0.001]. The WSC in the acceptable group was significantly different from the WSC in the borderline or the "poor" group (p = 0.034 - < 0.001).
The mean systolic blood pressures (SBP) was significantly different between the food consumption groups [F (2, 8610) = 7.21, p < 0.001]. The SBP in the acceptable group was significantly different from the SBP in the borderline group (p = 0.04). The SBP in the acceptable group was also significantly different from the SBP in the "poor" group (p < 0.001). However, the SBP of the borderline group did not significantly differ from the SBP of the "poor" group. The mean food consumption score (FCS) was significantly difference between the food consumption groups [F (2, 8610) = 11115.74, p < 0.001]. The FCS in the acceptable group was significantly different from the FCS in the borderline or the "poor" group (p < 0.001).
The mean moderate physical activity (MPA) days was significantly different between the food consumption groups [F (2, 8610) = 6.74, p = 0.001]. The MPA days in the borderline group were significantly different from the MPA days in the "poor" group (p = 0.011). The MPA days in the acceptable group was also significantly different from the MPA days in the "poor" group (p = 0.005). However, the MPA days of the borderline group did not significantly differ from the MPA days of the acceptable group. On the other hand, the mean vigorous physical activity (VPA) days was significantly different between the food consumption groups [F (2, 8610) = 3.99, p = 0.019]. The VPA days in the acceptable group were only significantly different from the VPA in the borderline group (p = 0.016) but not in the other groups.
The mean CES-D score was significantly different between the food consumption groups [F (2, 8610) = 16.60, p < 0.001]. The CES-D score in the borderline group was significantly different from the CES-D score in the "poor" group (p = 0.001). The CES-D score in the acceptable group was also significantly different from the CES-D score in the "poor" group (p < 0.001). However, the CES-D score of the acceptable group did not significantly differ from the CES-D score of the borderline group.
In 2007, the mean BMI was significantly difference between the food consumption groups [F (2, 8610) = 16.93, p < 0.001]. BMI in the acceptable group was significantly different from the BMI in the "poor" group. BMI in the borderline group was also significantly different from the BMI in the "poor" group (p < 0.001). However, the BMI of the acceptable group did not significantly differ from the BMI of the borderline group. The mean body shape index (BSI) was significantly difference between the food consumption groups [F (2, 8610) = 3.58, p < 0.001]. BSI in the acceptable group was significantly different from the BSI in the borderline group (p = 0.037). BSI in the acceptable group was also significantly different from the BSI in the "poor" group (p = 0.035). However, the BSI of the borderline group did not significantly differ from the BSI of the "poor" group. The mean WSC was significantly difference between the food consumption groups [F (2, 4658) = 18.83, p < 0.001]. The WSC in the acceptable group was significantly different from the WSC in the "poor" group (p < 0.001). WSC in the borderline group was also significantly different from the WSC in the "poor" group (p = 0.001). However, the WSC of the acceptable group did not significantly differ from the WSC of the borderline group.
The mean FCS was significantly difference between the food consumption groups [F (2, 8610) = 4313.01, p < 0.001]. The FCS in the acceptable group was significantly different from the FCS in the borderline or the "poor" group (p < 0.001). The mean MPA days was significantly different between the food consumption groups [F (2, 8610) = 6.60, p = 0.001]. The MPA days in the acceptable group were significantly different from the MPA days in the borderline group (p = 0.001). The MPA days in the acceptable group was also significantly different from the MPA days in the "poor" group (p = 0.004). However, the MPA days of the borderline group did not significantly differ from the MPA days of the "poor" group.
Further, the mean CES-D score was significantly different between the food consumption groups [F (2, 8610) = 20.95, p < 0.001]. The CES-D score in the borderline group were significantly different from the CES-D score in the "poor" group (p < 0.001). The CES-D score in the acceptable group was also significantly different from the CES-D score in the "poor" group (p < 0.001). However, the CES-D score of the acceptable group did not significantly differ from the CES-D score of the borderline group. The results of the Bonferroni post hoc test are in Table S1 and S2 in the supplemental materials.”
Comment 2:
Lines 142- 152, It is unclear why 5 versions of the adjusted models were made and why model 6 with all potential confounders was chosen.
Response 2:
Thank you for the comments on the logistic models. We used the six models in our logistic regression based on the previous study that the covariates (age, sex, education, marital status, geographical areas of living, smoking habits, days of doing physical activity, body mass index, blood pressures, diabetes, and cardiovascular diseases) might contribute to the association between food insecurity and the depressive symptoms. We gradually adjusted to see whether the association different after some of the covariates were included. After the six models were included in the analysis, we found the constant result that the association between food insecurity and depressive symptoms was positive and significant. Thus, we used the full adjustment to explore the stability of our findings in the post hoc analysis, as shown in Table 4.
Comment 3:
Table 3. There is no p-value for the variables. It's definitely better to compare OR instead of beta-coefficients.
Table 4. There is no p-value for the variables. It's definitely better to compare OR instead of beta-coefficients.
Response 3:
Thank you for your comments. We had addressed the p-value on Table 3 and Table 4 of our current manuscript on page 11-12 line 299 and line 304. Beta coefficient is an alternative way to interpret results depends on the research question [1,2]. The used of beta coefficient in the linear regression can be interpreted that a beta coefficient =1 means that if we change independent variable (iv) by 1, the expected value of dependent variable (dv) will go up by 1. On the other hand, the used of beta coefficient in the logistic regression can be interpreted that a beta coefficient =1 means that if we change iv by 1, the log of the odds that dv occurs will go up by 1. Thus, we used the exponentiated of beta coefficient in presenting the logistic regression value.
Comment 4:
In the logistics model, the results should be related to the reference level.
Response 4:
Thank you for your suggestion and comments. We had addressed the results section about the interpretation for logistic regression that should be related to the reference variable. The revised version can be addressed on page 6 line 270-279.
“Further, we used the logistic models to compare food security as represented by acceptable FCG and food insecurity as represented by borderline and poor FCG. The borderline group was positively associated with the depressive symptoms of both the unadjusted and adjusted models with exponentiated β-Coefficients: 1.13 (95% CIs: 1.06 to 1.21) to 1.18 (95% CIs: 1.10 to 1.26). The depressive symptoms of the borderline group will increase by 1.13 – 1.18 units in every one unit increase of the acceptable group. On the other hand, the "poor" group was also significantly positively associated with the depressive symptoms in both the unadjusted and adjusted models with exponentiated β-Coefficients: 1.17 (95% CIs: 1.07 to 1.27) to 1.22 (95% CIs: 1.12 to 1.33). The depressive symptoms of the "poor" group will increase by 1.17 – 1.22 units in every one unit increase of the acceptable group.”
Comment 5:
Correlation between depressive symptoms and food insecurity was observed only in the 40-49 years group. This relationship has not been interpreted precisely. The analysis are only partially reflected in the conclusions.
Response 5:
We had addressed the results section about the association between depressive symptoms and food insecurity was observed only in the 40-49 years group. The revised version can be addressed on page 6-7 line 282-289.
“The current study used a full adjustment model (model 6) in the analysis to examine the findings’ post hoc stability and decide whether the regression analysis different based on the age group. The “poor” food consumption group was significantly positively associated independently with depressive symptoms among respondents aged 40-49 years with exponentiated β-Coefficients: 1.24 (95% CIs: 1.08 to 1.42). The depressive symptoms of the "poor" food consumption group will increase by 1.24 units in every one-unit increase of the acceptable food consumption group only among respondents aged 40-49 years. The remaining age groups did not report food consumption score nor food consumption groups that were significantly associated with depressive symptoms.”
And in the conclusion section on page 14 line 392-394.
“To our knowledge, the present study results contribute to the evidence that food insecurity is positively significantly associated with depression symptoms among South-East Asian, in particular, Indonesian adults, and independently for people in their forties”.
The comments to the reviewers can also be addressed in the file entailed with it. Please see the attachment.
Reviewer 2 Report
Thank you for allowing me to review your study.
While the concept is interesting and the idea worth pursuing, as it stands it is very difficult to read and understand what it is that you actually did. You call FCD continuous but present is ordinally. You talk about correlations but you mean associations and coefficients I think. Your models look ordinal but you talk about logistic (so are you using low as yes no and then medium as yes no etc??. You say that you have repeated measures but you are not using repeated measures analysis. You have to do this with repeated measures.
So I have to say I cannot adequately assess this paper as it is currently written. You really have to be clearer about what you did. I suggest that you have the manuscript edited for English usage as well.
Author Response
REVIEWER 2
Comment 1:
Thank you for allowing me to review your study. While the concept is interesting and the idea worth pursuing, as it stands it is very difficult to read and understand what it is that you actually did.
You call FCS continuous but present is ordinally.
Response 1:
Thank you for the comments. We added the information about the food consumption analysis that resulted in food consumption score (FCS) on the methods. We added in the statistical analysis on page 3 line 107-110.
“a total from each food group called food consumption score (FCS) then categorized based on the cutoffs of three food consumption groups (FCG). The FCS is continuous data, while the FCG is categorical data from the categorization of FCS.”
Comment 2:
You talk about correlations but you mean associations and coefficients I think. Your models look ordinal but you talk about logistic (so are you using low as yes no and then medium as yes no etc.??
Response 2:
Thank you for your constructive comments and suggestion. We revised the current manuscript from the “correlation” to association. The current revision can be addressed on the title as well as on the body of the manuscripts.
“Association between Depressive Symptoms and Food Insecurity among Indonesian Adults: Results from the 2007 - 2014 Indonesia Family Life Survey”.
In the statistical analysis we used two regression models, the linear model for the continuous data, and the logistic model for the categorical data. The continuous data, for example, the food consumption score (FCS). The categorical data, such as food consumption group, which is a categorization of the food consumption score. There were three groups in the FCG, “Poor” (if the FCS value is ≤ 21), “Borderline” (if the FCS value 21 - 35), and “Acceptable” (if the value ≥ 35).
Comment 3:
You say that you have repeated measures but you are not using repeated measures analysis.
You have to do this with repeated measures.
Response 3:
Thank you for the comments. We used the secondary data from the IFLS4 and IFLS5. These datasets were using the same respondents that have been followed from the first IFLS (IFLS1) that initiated in 1993. The current study was included the respondents who have no missing data in both IFLS4 and IFLS5 for the FFQ data, CES-D score, age, sex, blood pressures, and the anthropometric data except for the waist circumference that only measured for the respondents aged above or equal of 40 years. Therefore, the data was repeated measured and we used the generalized estimating equation test with linear and logistic model to analyzed this data. GEE is widely used to analyzed such a repeated measurement data [3-6].
Comment 4:
So, I have to say I cannot adequately assess this paper as it is currently written. You really have to be clearer about what you did. I suggest that you have the manuscript edited for English usage as well.
Response 4:
Thank you for your suggestion. We had revised the current manuscript and we send it for the English editing as well.
The response to the reviewer also can be addressed in the file entailed with it. Please see the attachment.
Reviewer 3 Report
Manuscript presented by Isaura et al. concerns a very important and at the same time complex topic of depressive symptoms and lack of food security. These are two public health problems in developing countries that, along with obesity-related non-communicable diseases, significantly affect people's quality of life.
This is a large and interesting study but requires more details
The introduction is a cursory introduction to the topic of work. A change of structure should be considered to provide the reader with the main rational and justification for this study. It is unclear what is new in this study, what it adds and what would contribute to current evidence.
I propose to present the exact purpose of the study and the research hypothesis.
I suggest adding some more information about the selection of participants in chapter/section: Data Source and Respondents. Perhaps it would be better to present the recruitment and inclusion of participants as a scheme (a sampling block scheme is recommended), including on/off criteria for the study.
The methodology should be more clarified.
Was the FFQ questionnaire used validated in earlier studies? If so, please provide the reference.
Have anthropometric measurements been carried out, in what conditions and by whom? - more data should be provided.
The sentence (line 100-101) "Further, the author then rebased the score from 10 – 40 to 0 – 30, so the score of ≥ 10 was considered as the cut-off point for respondents having depressive symptoms" should be clarified.
Fragment - lines 122-127 - requires explanation
No explanation of all abbreviations used in the work, e.g. what is FCS?
The chapter/section Results requires more consideration.
Whether the two datasets (IFLS 4 and 5) connection was correct - some variables were statistically significantly different.
The discussion requires refinement, highlighting the strengths of the study and giving what's novelty.
Author Response
REVIEWER 3
Comment 1:
Manuscript presented by Isaura et al. concerns a very important and at the same time complex topic of depressive symptoms and lack of food security. These are two public health problems in developing countries that, along with obesity-related non communicable diseases, significantly affect people's quality of life.
This is a large and interesting study but requires more details.
The introduction is a cursory introduction to the topic of work. A change of structure should be considered to provide the reader with the main rational and justification for this study. It is unclear what is new in this study, what it adds and what would contribute to current evidence.
I propose to present the exact purpose of the study and the research hypothesis.
Response 1:
Thank you for the constructive comments and suggestions. We added the information in the manuscript on page 14 line 391-394.
“The present study concerns a very important and at the same time, a complex topic of depressive symptoms and lack of food security. These are two public health problems in developing countries that, along with obesity-related non-communicable diseases, significantly affect people's quality of life.”
We also addressed the introduction section on page 2 line 48-80.
Depression is one of the public health problems, which is associated with adverse mental health such as suicidal ideation and mortality [7]. Depression is defined as a wide range of mental health problems associated with the negative effect presence, low mood, and emotional, cognitive, physical, and behavioral symptoms [8]. On the other hand, depression is a pervasive mental disorder in the world that affects all of the ages [9]. In 2012, the effect of depression on people was estimated to reach about 350 million in the world [9]. Further, the global population with depression is estimated to be 4.4% in 2015, while Indonesia's national prevalence rate for people having depressive symptoms is 3.7% [10]. The occurrence of the depression or depressive symptoms can come in the episodic sequences [8]. Some unwanted life events (e.g., the loss of a loved one or separation in a relationship), or living with poverty, being unemployed, having a physical illness, and drug and alcohol use-related problems increase the risk of depression or having depressive symptoms [10,11]. Furthermore, an adult who is unemployed or living with poverty is also associated with food insecurity because of the financial resources limit for acquiring food and managing their diet [12].
Food insecurity defines as a condition that a person has limited or uncertain availability or access to nutritionally adequate, culturally relevant, and safe foods [12]. Moreover, food insecurity was found to be associated with chronic diseases [13,14]. The former researchers suggested that may be chronic diseases are a contributing factor in the association between food insecurity and depression among the elderly [15-17]. Food insecure people are prone to consume an energy-dense and less diverse diet, which eventually results in overweight and obesity, higher risk of hypertension, diabetes, and cardiovascular diseases [13,14,18,19]. On the other hand, Seligman and Schillinger suggested that there is a trade-off between providing food and buying medicines in the association between food insecurity, chronic diseases, and depressive symptoms [18]. Not only the association between food insecurity and depression or depressive symptoms is rather vague among adults, but both food insecurity and depression or depressive symptoms can also affect people, women in particular who live in high-income or low-middle income countries [20]. Therefore, the authors were used different methods and study designs to explore further and evaluate whether some specific age group modified the association between food insecurity and depressive symptoms among Indonesian adults. In this study, we used the repeated measurement data to assess the association between food insecurity and depressive symptoms in adults, both in all ages and the various age groups. Besides, we observed depression or depressive symptoms as both predictor and outcome and used different food insecurity assessment.
Comment 2:
I suggest adding some more information about the selection of participants in chapter/section: Data Source and Respondents.
Perhaps it would be better to present the recruitment and inclusion of participants as a scheme (a sampling block scheme is recommended), including on/off criteria for the study.
Response 2:
We added the information about the selection of participants in chapter/section: Data Source and Respondents on page 2 line 85-92.
“In 2007 data, the total respondent was 29059 people while in the 2014 data total respondent was 34464 people (aged 0 – 80+). For this study, we included adults’ respondents who are 18-65 years old. We included the same respondents from the year 2007 and the year 2014. The respondents who have the completed data on dietary, physical activity, anthropometric, sociodemographic, blood pressures and depressive symptoms were further analyzed. We excluded the respondents who were pregnant or breastfeeding, having disabilities, and who had diagnosed with cancer to minimize the sampling bias. Further, we included only respondents who have no missing data in both year 2007 and 2014. After the inclusion criteria applied, therefore, the 8613 respondents were included in this study.”
Comment 3:
The methodology should be more clarified. Was the FFQ questionnaire used validated in earlier studies? If so, please provide the reference.
Response 3:
The FFQ was developed and was used in earlier studies [3,21-30].
Comment 4:
Have anthropometric measurements been carried out, in what conditions and by whom? - more data should be provided.
Response 4:
Thank you for the constructive comments. We added the information in the measurement of the covariates section, on page 3 line 126-132.
“Body mass index (in kg/m2) was classified into four groups (< 18.5, 18.5-25.0, 25.1-27.0, and >27.0) were used [31]. In addition, a measurement of waist circumference was used for adults aged ≥ 40 years. The abdominal obesity was determined by respondents’ waist circumference (WC) > 90 cm (men) and > 80 cm (women) [32]. Hypertension was defined as systolic blood pressure (SBP) is ≥ 140 mmHg or diastolic blood pressure (DBP) ≥ 90 mmHg) [33]. The trained nurses performed the anthropometric and blood pressure measurements. For the blood pressure measurements, the respondents were measured twice, before and during the interview, in the seated position.”
Comment 5:
The sentence (line 100-101) "Further, the author then rebased the score from 10 – 40 to 0 – 30, so the score of ≥ 10 was considered as the cut-off point for respondents having “depressive symptoms" should be clarified.
Response 5:
We added the information in the section : measurement od depressive symptoms on page 3 line 115 – 124.
“Depressive symptoms were assessed using self-reported ten items Center for Epidemiologic Studies Depression (CES-D) questionnaire. The CES-D is widely used to assess the depressive symptoms on adults [34,35]. The 10-CES-D answers were in the form of four scales, such as “Rarely or no (≤ 1 day)”, “Some days (1-2 days)”, “Occasionally (3-4 days)”, “Most of the time (5-7 days)”. The score of each scales answer was from zero (rarely or no) to four (most of the time). We then summarized the score of 10-CES-D with the lowest score was 10, while the highest score was 40. Since the score was from 10 to 40, therefore, the score was rebased to zero to 30 with the highest score refers to the most depressive symptomatology [34]. Previous researches suggested the cutoff point for depression or having depressive symptoms was set to the score of ≥ 10 [34,36,37]. Therefore, respondents were defined as depression or having depressive symptoms if their CES-D score ≥ 10.”
Comment 6:
Fragment - lines 122-127 - requires explanation
Response 6:
We had addressed the information in the measurement of the covariates section, on page 4 line 145-153.
“In addition to the covariates variables, we consider adjusting the respondents’ chronic diseases. Therefore, this study was used cardiovascular diseases and type 2 diabetes as an adjustment variable in the model. Respondents answered the self-reported question of whether any paramedics ever informed them that they had type 2 diabetes. The respondents also answered the self-reported question of whether any paramedics ever informed them that they had stroke/heart attack, coronary heart disease, angina, or other heart problems. The authors defined cardiovascular disease as a combination of heart diseases and stroke events [21,38]. If the respondents reported any of the chronic diseases (i.e., diabetes, cardiovascular diseases), then they were asked when their chronic disease first diagnosed.”
Comment 7:
No explanation of all abbreviations used in the work, e.g. what is FCS?
Response 7:
We added the information about the abbreviations e.g., FCS in the section: measurements of food insecurity on page 3 line 108-113.
“…a total from each food group called food consumption score (FCS) then categorized based on the cutoffs of three food consumption groups (FCG). The FCS is continuous data, while the FCG is categorical data from the categorization of FCS. The three FCGs are "Poor" if the FCS value is less than 21, "Borderline" if the FCS value from 21 to 35 and "Acceptable" if the value more than 35 [39]. Finally, this study was defined food-insecure people who are in the "poor and borderline" group of FCGs while food-secure people who are in the "acceptable" group of FCG [40,41].”
Comment 8:
The chapter/section Results requires more consideration.
Response 8:
Thank you for your suggestions. We had addressed the section results with additional information related the interpretation of the tables and also the interpretation of some supplemental tables that supporting the main table in the manuscript. The revision can be addressed on page 5-12 on the current version of manuscript.
Comment 9:
Whether the two datasets (IFLS 4 and 5) connection was correct - some variables were statistically significantly different.
Response 9:
The datasets of the IFLS4 and IFLS5 was two datasets that taken in the different year. The IFLS4 was collected in year 2007, and IFLS5 was collected in year 2014. The respondents that included in the latest datasets (IFLS5 data) was some parts the same respondents as the ones in IFLS4. Some of the same respondents has been followed for all waves of IFLS. However, to represent the latest trend of the Indonesia, therefore, we used only the two latest IFLS datasets (IFLS4 and IFLS5). Thus, some variables can show the significantly different because the gaps between the datasets was 7-year.
Comment 10:
The discussion requires refinement, highlighting the strengths of the study and giving what's novelty.
Response 10:
Thank you for the constructive comments and suggestions. We added the information in the discussion sections, as well as highlighting the strengths of the study and giving what is novelty. The revision can be addressed on page 13 line 311-394.
Please also see the attachment for more information of the response to the reviewers. Thank you.
Round 2
Reviewer 1 Report
Dear Authors,
Thanks to the authors for including comments in manuscript.
Author Response
Thank you to the reviewers for giving us constructive and helpful suggestions as well as comments on our manuscript.

Reviewer 2 Report
Thank you for the opportunity to once again review your paper. However you have not addressed two of my major concerns. The first is easily done - that is have your manuscript edited by a native English speaker because as it stands there are still incomplete sentences, phrases and sentences that do not make sense.
However the second issue is with the analysis. As a general rule it is bad practice to categorise when it is not necessary so if you have age as a continous variable then leave it as such unless you have evidence to show that a certain age group or range is associated with your outcome.
I also don't know why you do not use a repeated measures analysis when you state you are doing repeated measures - GEE is not correct if you are looking at the same person over two different points in time. If I am correct, you are combining two datasets using the same person twice - so looking at changes over time, then you have to use a time interaction with your variable of interest - in this case food insecurity. There are many statistical articles that show this to be the case. If however you are not using the same person over time, then what you are doing is ok but why use GEE and not standard regression.
You can use regression for repeated measures as well... if you look up repeated measures in STATA you will find references to this.
The number of models is excessive and unnecessary - it is confusing for the reader to understand why you are using so many. What point are you trying to make?
You can make your paper much more interesting and easy to understand if you simply what you do and use the correct stats to do so. Maybe you have but as you have written it up I just cannot judge.
Author Response
§ REVIEWER 2
Comment 1:
Thank you for the opportunity to once again review your paper.
However, you have not addressed two of my major concerns.
The first is easily done - that is have your manuscript edited by a native English speaker because as it stands there are still incomplete sentences, phrases and sentences that do not make sense.
Response 1:
Thank you for the comments and suggestions. We had sent our revised manuscript to the MDPI English editing, and we received the certificate about it. However, there was some error in the submission process, and we had contacted the assistant editor to help us fix this issue. Thus, the current version of the manuscript is including the edited version from the MDPI English editing service.
Comment 2:
However the second issue is with the analysis. As a general rule it is bad practice to categorise when it is not necessary so if you have age as a continous variable then leave it as such unless you have evidence to show that a certain age group or range is associated with your outcome.
Response 2:
Based on the former researchers' findings [1-4], we consider using age categories in our study in addition to the continuous form. In the statistical analysis, we used variable age as both continuous and categorical form. We used age as the continuous variable during the linear regression test and in the model 2 - 3 of logistic regression as shown in Table 3. Since we aimed to explore that age group may modify the association between food insecurity and depressive symptoms, therefore, we used age as categorical variable. We used age as a categorical variable as a piece of additional information in Table 1 and Table 2, as well as in the analysis of the post hoc test, as shown in Table 4.
Comment 3:
I also don't know why you do not use a repeated measures analysis when you state you are doing repeated measures - GEE is not correct if you are looking at the same person over two different points in time. If I am correct, you are combining two datasets using the same person twice - so looking at changes over time, then you have to use a time interaction with your variable of interest - in this case food insecurity. There are many statistical articles that show this to be the case. If however you are not using the same person over time, then what you are doing is ok but why use GEE and not standard regression.
You can use regression for repeated measures as well... if you look up repeated measures in STATA you will find references to this.
Response 3:
We had addressed the reason why we used GEE in our study. We had revised the GEE explanation in the statistical analysis part on page 4 line 159-164.
“Since the data in this study were repeated measurement data from the same respondents throughout the 7-year follow-up period, the authors used regression models with the generalized estimating equation (GEE) method [5]. The GEE is a statistical approach generally used in the analysis of longitudinal data or repeated measurements [6-11], with the primary advantage of it accounts for the within-adults variation [12].”
In addition, we also did the repeated measurement using the standard regression as suggested by the reviewer. Table R1 shows the repeated measures using the standard logistic regression. The association between food insecurity and depressive symptoms are significantly positive. Food-insecure people (poor FCG) are 1.175 times more likely to experience depressive symptoms rather than food-secure people (acceptable FCG). The results of the current study using GEE is consistent with these results. However, the result in table R1 cannot describe the correlations between binary outcomes across time within the same individual. Therefore, we consider the GEE test that suits with our study.
Table R 1. Repeated Regression Results
FCG |
Model 1 |
Model 2 |
Model 3 |
|||
OR (95% CI) |
p-value |
OR (95% CI) |
p-value |
OR (95% CI) |
p-value |
|
2007 |
||||||
Acceptable |
1 (Ref.) |
1 (Ref.) |
|
1 (Ref.) |
|
|
Borderline* |
0.869 |
0.364 |
0.854 |
0.308 |
0.835 |
0.248 |
Poor* |
1.155 |
0.271 |
1.150 |
0.288 |
1.101 |
0.470 |
2014 |
||||||
Acceptable |
1 (Ref.) |
1 (Ref.) |
|
1 (Ref.) |
|
|
Borderline* |
1.060 |
0.385 |
1.037 |
0.594 |
1.038 |
0.585 |
Poor* |
1.207 |
0.003 |
1.193 |
0.006 |
1.175 |
0.014 |
Notes: FCG, food consumption group. Model 1, unadjusted model. Model 2, adjusted by age and sex. Model 3, model 2 with additional adjustment for level of education, marital status, and geographical areas of living, smoking habit status, physical activity days, blood pressure values, body mass index, and included diabetes and cardiovascular diseases. Depressive symptoms were defined as CES-D 10 score ≥10.
Comment 4:
The number of models is excessive and unnecessary - it is confusing for the reader to understand why you are using so many. What point are you trying to make?
You can make your paper much more interesting and easy to understand if you simply what you do and use the correct stats to do so. Maybe you have but as you have written it up I just cannot judge.
Response 4:
We had revised the statistical analysis of the logistic regression models as well as the results tables. We had removed the model 3 – 5, and change the model 6 as model 3 in the current version of the manuscript. The revised version can be addressed on page 4 lines 170-175 in the main text and table 3 on page 9.

Reviewer 3 Report
The previous concerns expressed by this reviewer were addressed by the authors in the revised manuscript. However, the exact purpose of the thesis and the research hypothesis have still not been presented.
The description of the results is too long, and there are repeats of data from the tables.
Despite this, the authors have made a lot of effort to improve the manuscript.
I can now support its application.
Author Response
§ REVIEWER 3
Comment 1:
The previous concerns expressed by this reviewer were addressed by the authors in the revised manuscript. However, the exact purpose of the thesis and the research hypothesis have still not been presented.
Response 1:
Thank you for the helpful comments and constructive suggestions for our manuscript. The purpose of this study was to explore the association between food insecurity and depressive symptoms among adults. The previous study results that suggested age group may modify the association, so we consider further analyze it in our study. The current study’s research hypothesis was food insecurity was positively associated with depression, and some age groups may modify this relationship.
These statements can be addressed in the current version of the manuscript on page 2 line 73-79.
“Therefore, in this study, we used different methods and study designs to further explore and evaluate whether specific age groups modified the association between food insecurity and depressive symptoms among Indonesian adults. We used repeated measurement data to assess the association between food insecurity and depressive symptoms in adults, both in all ages and in various age groups. In addition, we observed depression or depressive symptoms as both predictor and outcome, and used different food insecurity assessments.”
Comment 2:
The description of the results is too long, and there are repeats of data from the tables.
Despite this, the authors have made a lot of effort to improve the manuscript.
I can now support its application.
Response 2:
We had revised our “results” part and removed the description of the Supplemental Tables from the main text into the Supplemental Materials together with Table S1 and S2. The revised version can be addressed on page 5 lines 182-228 in the main text and below table S1-S2 in the Supplemental Material.
